# Social Vulnerability of Brazilian Metropolitan Schools and Teachers’ Absence from Work Due to Vocal and Psychological Symptoms: A Multilevel Analysis

**DOI:** 10.3390/ijerph20042972

**Published:** 2023-02-08

**Authors:** Adriane Mesquita de Medeiros, Mariana Fernandes Lobo, Marcel de Toledo Vieira, Lia Duarte, João Paulo Monteiro Carvalho, Ana Cláudia Teodoro, Rafael Moreira Claro, Nayara Ribeiro Gomes, Alberto Freitas

**Affiliations:** 1Postgraduate Program in Speech-Language Sciences, Federal University of Minas Gerais, Belo Horizonte 30130-100, Brazil; 2Postgraduate Program in Public Health, Federal University of Minas Gerais, Belo Horizonte 30130-100, Brazil; 3CINTESIS@RISE, MEDCIDS, Faculty of Medicine of the University of Porto, 4200-450 Porto, Portugal; 4Department of Statistics and Graduate Program in Economics, Federal University of Juiz de Fora, Juiz de Fora 36036-900, Brazil; 5Institute of Earth Sciences, FCUP Pole, Rua do Campo Alegre, 4169-007 Porto, Portugal; 6Department of Geosciences, Environment and Spatial Planning, Faculty of Sciences of the University of Porto (FCUP), Rua do Campo Alegre, 4169-007 Porto, Portugal

**Keywords:** voice disorders, mental health, social vulnerability index, absenteeism, teachers, multilevel analysis, GIS analysis

## Abstract

Teachers’ voices and psychological symptoms are the main reasons for absence from work. The objectives of this study were: (i) to spatially represent, through a webGIS, the standardized rates of teachers’ absences due to voice (outcome 1) and psychological symptoms (outcome 2) in each Brazilian Federative Unit (FU = 26 states plus Federal District) and (ii) to analyze the relationship between each national outcome rate and the Social Vulnerability Index (SVI) of the municipality where urban schools are located, adjusted for sex, age, and working conditions. This cross-sectional study comprised 4979 randomly sampled teachers working in basic education urban schools, of which 83.3% are women. The national absence rates were 17.25% for voice symptoms and 14.93% for psychological symptoms. The rates, SVI, and school locations in the 27 FUs are dynamically visualized in webGIS. The multilevel multivariate logistic regression model showed a positive association between voice outcome and high/very high SVI (OR = 1.05 [1.03; 1.07]), whereas psychological symptoms were negatively associated with high/very high SVI (OR = 0.86 [0.85 0.88]) and positively associated with intermediate SVI (OR = 1.15 [1.13; 1.16]), in contrast with low/very low SVI. Being a woman (voice: OR = 1.36 [1.35; 1.38]; psychological: 1.22 [1.21; 1.24]) and working in schools with various precarious conditions (17 variables) increased the odds of being absent due to voice and psychological symptoms. The results confirm the need for investments to improve working conditions in schools.

## 1. Introduction

Employees that are recurrently on sick leave may come to have increasingly longer absences from work overtime [1]. Chronic health conditions increase the likelihood of sick leaves, raising the costs to employers and the economy, and consequently being disadvantageous to employees [2]. Due to absenteeism, they may lose income, promotion, or earlier retirement opportunities and can be at greater risk of being dismissed [3]. The heterogeneous behavior of employees who are absent from work must be considered [4]. In the specific case of teachers, their absences from work may also lead to poorer interpersonal relationships at school and work overload for other teachers who carry out the activities of the absent ones [3,5]. Moreover, it can impair students’ learning and weaken the school community’s effectiveness [6,7].

Sick absenteeism from work is an indicator of the employee’s health status and the effects of the environment and organizational work factors on their health [8]. It has been proven that employees can be simultaneously exposed to multiple risk factors for absence from work [9]. Most Brazilian teachers’ absences were due to voice and psychological symptoms [9,10,11]. Having three or more medical visits for chronic diseases (cardiometabolic, psychological, orthopedic, respiratory, gastrointestinal, nervous system diseases, and cancer) increased the odds of Brazilian teachers being on sick leave [11].

Studies on stress and burnout syndrome stand out in the literature on the relationship between faculty health and working conditions [12]. The analysis of long-lasting sick absenteeism due to burnout verified its relationship with teachers’ depersonalization i.e., when professionals start treating students, workmates, and the organization distantly and impersonally [13]. Another study on childhood education teachers showed that increased depressive symptoms were further associated with more days absent over the school year. Both anxiety and depressive symptoms were associated with less job satisfaction [14].

Besides mental health problems, teachers’ voice symptoms hinder them from carrying out work tasks, as their voice is greatly required in the classroom [15]. Teachers’ more severe voice symptoms proved to be related to absenteeism due to voice disorders, health service use, and decreased productivity at work [16]. Voice-related sick leaves are mostly short [17] and more prevalent in teachers who are absent from work also due to psychological symptoms [18,19].

Schools with fewer resources, precarious working conditions, and whose children have family problems and live in poorer neighborhoods can increase teachers’ stress and affect their health, thus making them sick and causing them to be absent from work [6]. The socio-professional neighborhood was related to burnout in French teachers [20]. In Brazil, teachers’ activities are increasingly complex as school demands intensify. Hence, when schools are in a poorer and needy context, greater demands are posed to them and their teachers [21].

Brazil has striking socioeconomic inequality, which directly or indirectly affects most of the population. The social vulnerability index (SVI), which is periodically calculated regarding the various regions of Brazil, reveals the absence or insufficiency of greater assets, such as infrastructure, human capital, employment, and income for inhabitants in municipalities [22]. SVI identifies the heterogeneity of Brazilian metropolitan populations in terms of vulnerability to social exclusion.

Geographical Information Systems (GIS) have been widely used in the context of health conditions providing tools to store, manage, and analyze data that requires spatial distribution [23,24,25,26,27,28]. In the last decades, webGIS applications have been widely developed to dynamically represent variables in the health context. For instance, Kienberger et al. (2013) [29] developed a webGIS to spatially visualize vulnerabilities to dengue fever in Cali (Colombia). Khoshabi et al. (2016) [30] developed an interactive webGIS tool to dynamically visualize and analyze cancer data and environmental parameters for a database in Iran. This study also implemented different forms to visualize cancer occurrence in the form of bar and pie charts. Regarding geospatial data and school mapping assessment, Agrawal and Gupta (2020) [31] developed a webGIS framework to reduce costs in education and promote technological development for primary education in rural areas, especially in developing countries like India. Duarte et al. (2021) [32] developed a GIS open-source application in QGIS open-source software to assess geographical patterns and trends of health data in Portugal, as well as a webGIS to visualize the spatial assessment of health care quality indicators. However, no webGIS was found in the consulted literature that allows the spatial representation of SVI per school and related teachers’ leaves of absence. 

Therefore, this study is justified by the scarcity of scientific evidence of the relationship between the social vulnerability of school surroundings and teachers’ leaves of absence. Thus, the objectives of this study were: (i) firstly, to spatially represent, through a webGIS, the teachers’ absence rates due to voice and psychological symptoms in the 26 Brazilian states and the Federal District (27 Federative Units [FU]) and (ii) secondly, to analyze the relationship between the national rate of each outcome and the SVI of the municipality where Brazilian basic education metropolitan schools are located, adjusted for sex, age, and working conditions.

## 2. Materials and Methods

This cross-sectional study, based on information from the National Survey of Teachers’ Working Conditions and Absences in Brazilian Basic Education Schools (Educatel), was conducted from October 2015 to March 2016. All subjects gave their informed consent for inclusion before they participated in the study. The study was conducted in accordance with the Declaration of Helsinki, and the protocol was approved by the Ethics Committee under no. 1305863.

### 2.1. Study Population and Data Collection

The Educatel sample comprised 6510 teachers of both sexes recruited based on a stratified sampling strategy developed with data from the 2014 School Census. Teachers were initially stratified per Regions of Brazil (North, Northeast, Central-West, Southeast, and South), urban and rural areas, teachers’ sex and age, employment relationship (civil servant, permanent, stable, temporary, labor-law contract), school’s administrative status (private, state, municipal, or federal), and the level of education in which the teachers worked (from preschool to high school). Next, the teachers were randomly selected within strata. The sampling process was detailed by Vieira et al. [33]. Teachers were interviewed by phone, after the previous contact with a school assistant to schedule the interview. The questionnaire was evaluated, and the interviewers were trained in the pilot stage. Secondary administrative and school data were also obtained from the 2014 School Census. 

This study included information from the 2010 SVI of Brazilian municipalities where participating teachers’ urban schools were located. SVI is public domain (http://ivs.ipea.gov.br/index.php/pt/ accessed on 10 January 2022) and available from the Brazilian Institute of Applied Economic Research (Ipea) with data from the demographic census conducted every 10 years by the Brazilian Institute of Geography and Statistics (IBGE). There are no data on rural SVI because one of the urban infrastructure indicators is living in an urban household with no waste collection. Hence, only urban areas were calculated [22].

This study sample comprised 4979 teachers, representative of classroom teachers who work in Brazilian metropolitan schools at the time of data collection. Those who worked in rural schools were excluded (15.929%) since there is no calculation of the SVI for rural regions. 

### 2.2. Measurements

The outcome variables were defined after teachers affirmatively answered the following questions: “In the last 12 months, were you absent from work for at least 1 day (regardless of the reason)?”, “Were you absent from work because of health issues?”, “What was the health issue?”. Various reasons to be absent from work were investigated; the following ones were of interest to this study: having or not having voice symptoms (e.g., hoarseness, voicelessness) and psychological symptoms (e.g., depression, stress, anxiety).

The explanatory variable was SVI with three subindices (Infrastructure SVI, Human Capital SVI, Income and Employment SVI), based on 16 indicators that help understand people’s living conditions, identifying the ones at social risk and social vulnerability [22].

Urban infrastructure SVI assesses the extent to which people have basic sanitation and urban mobility available. Human Capital SVI analyzes structural aspects and assets that determine the current and future perspective of educational and health inclusion, such as child mortality, illiteracy, teenage motherhood (10 to 17 years old), and so forth. Lastly, Income and Employment SVI is based on unemployment rates, family monthly income lower than or equal to half minimum wage, low-income families who depend on older adults’ income, child labor, and informal work without having finished middle school [22].

SVI is calculated by the mean of the three subindices, tabulated with their respective weights. Each indicator required maximum and minimum parameters, standardizing with values from zero (the ideal or desirable condition) to one (greater social vulnerability) [22].

This study considered the social vulnerability ranges for municipalities shown in Table 1, grouping SVI into three categories: (1) very low and low, (2) intermediate, (3) high, and very high. 

The following covariables were addressed: more than 40-h working week (“Considering all the schools where you currently work as a teacher, what is your weekly workload?”); employment time (“How long have you been working in basic education?”); school administration (secondary data from the school census); income from school in minimum wages (“What is your monthly wage earned from working at this school?”); working in more than one school (“Do you work at more than one school?”); having another occupation (“Do you have any paid activity in a sector other than education?”); perception of intense noise at school (“How often is the noise at work so loud that you have to raise your voice to talk to someone else?”); student indiscipline (“How often is your work environment hectic because of students’ indiscipline?”); commute time (“Approximately how much time do you spend commuting from home to work every day—round trip?”); experience with verbal violence at school (“In the last 12 months, have you suffered verbal violence from students?”) and physical violence at school (“In the last 12 months, have you suffered physical violence from students?”); school size by the number of teachers (secondary data from the school census); excessive workload (“Does your job demand too much of you?”); few opportunities to learn new things (“Do you have the opportunity to learn new things in your work?”); insufficient time to finish tasks (“Do you have enough time to complete all your work tasks?”); limited autonomy (“Does this school provide opportunities for staff to actively participate in the decisions that are made?”); and low social support (six questions: “Is your work setting calm and pleasant?”, “At work, does everyone get along well with each other?”, “Can you count on support from your coworkers?”, “If you’re not having a good day, do your coworkers understand?”, “At work, do you get along well with your superiors?”, and “Do you enjoy working with the other teachers?”). The answers to the six questions in the Job Stress Scale (JSS) [34] were summed (answering “often” scored 1 point, “sometimes” scored 2, “rarely” scored 3, and “never or almost never” scored 4). Scores greater than 7 (50th percentile) were defined as not having social support at school [34]. The answer options for each question were included in Table 2.

### 2.3. Statistical Analysis

The multivariate analysis consisted of a multilevel logistic regression model, in which the FU effects were modeled with a normally distributed random intercept (details of the models in the Appendix A). This type of model accounts for the potential correlation of outcomes of schoolteachers clustered within FU, modeling the variation of teachers’ outcomes within and between FUs with a random effect [35,36]. This approach not only prevents spurious estimates of fixed effects but also the estimation of FU’s specific effects. The fixed effects of the regression models included the explanatory variable (SVI) that characterizes municipalities where schools are located as well as teacher-level control variables (age, sex, and working conditions, as in Section 2.2). The models were implemented in R software, v.4.1.3, with RStudio, v.2022.02.0, using the *glmer* function in the *lmer* package.

The overall crude national absence rate due to voice and psychological symptoms was calculated taking into account the data’s complex sampling. Voice and psychological absence adjusted for each FU were obtained through indirect standardization controlling for sex, age, SVI, and working conditions. Adjusted rates result from the ratio between predicted odds (P), which consider the specific effect of each school, and the expected odds (E) of the outcomes in each FU. This approach is an adaptation of the observed (O)-to-expected odds ratio (O/E), with the advantage of stabilizing the ratio in FUs with smaller samples, shrinking estimates towards the mean. The P/E ratios for each FU were then multiplied by the national absence rate, which was calculated according to the complex study sample design [35]. Absence rates of FUs with ratios = 1 are equivalent to the national rate.

The fixed effects were described with relative frequencies and characterized by adjusted odds ratios and their respective 95% confidence intervals and *p*-values. The Intraclass Correlation Coefficient (ICC) was calculated to quantify the contextual effect of the FU, which measures the proportion of observed variation in the odds of being absent from work, ascribable to differences across FUs (while controlling for fixed effects). ICC values range from 0 to 1, in which an ICC = 0 indicates no influence from the FUs on the teacher outcomes, and an ICC = 1 indicates that teachers’ outcomes are completely determined by their FUs [36]. The Median Odds Ratio (MOR) quantified the heterogeneity between FUs [37], and the *c-statistic* was used to assess the predictive capacity of the model [38].

We also assessed the discriminative ability of the multilevel models with c-statistic by comparing the area under the curve of a model with age and sex variables only and random effects to the multilevel model described above. A value of 1 indicates that the model perfectly discriminates against teachers who were absent from work (versus not absent) as a function of teachers’ predicted probabilities, whereas a value of 0.5 represents random discrimination.

### 2.4. WebGIS

A WebGIS was developed to spatially represent all acquired and processed data: SVI of the municipalities, school location based on the geographical coordinates (in World Geodetic System 1984 [WGS84]), FU, absence rates for voice symptoms, and absence rates for psychological symptoms. 

The geographical coordinates (longitude and latitude) of each participating teacher’s school were obtained from *gpsvisualizer* (https://www.gpsvisualizer.com/, accessed on 10 July 2022) to locate it based on its full address (street, number, neighborhood, city, state, and country). The Plus Code was identified with Google Maps if it lacked geographic coordinate information. 

The WebGIS used a python backend framework called *Flask* (https://flask.palletsprojects.com/en/2.1.x/, accessed on 1 November 2022), which makes it possible to respond to the user’s requests (data). This procedure connects with the open-source database management system PostgreSQL (https://www.postgresql.org/, accessed on 1 November 2022) where all data is stored, using the python *psycopg2* function, which connects to the database and allows for queries. The front end was developed with Hyper Text Markup Language (HTML), Cascading Style Sheets (CSS), and JavaScript programming languages. The data were rendered with a JavaScript library called *Mapbox GL JS*, a JavaScript library for vector maps on the Web (https://www.mapbox.com/, accessed on 1 November 2022). 

The WebGIS has a menu on the left bar to activate/deactivate the layers (one for each dataset), visualize SVI values associated with each municipality through a popup, and enable 3D visualization. These include the bars to analyze and compare teachers’ rates of absence from work due to vocal and psychological symptoms for each FU in Brazil (27 states and the federal district).

The geospatial information on Brazilian municipalities was freely obtained from Natural Earth (https://www.naturalearthdata.com/, accessed on 1 April 2022), and the polygons related to the FUs were obtained by applying a dissolve to the municipalities using the *dissolve* algorithm from QGIS. The layer referring to school location has the geographic coordinates originally organized in an Excel spreadsheet and then uploaded to PostgreSQL. Lastly, the locations were imported to WebGIS as points, creating a legend for each layer. The WebGIS also has standard tools, such as zoom in, zoom out, pan, and so on.

## 3. Results

The developed WebGIS allows us to visualize the location of the schools, the municipal SVI, and the absence rates for voice and psychological symptoms separately (Figure 1, Figure 2 and Figure 3) or dynamically (e.g., with the possibility of overlaying or zooming in data). Please see Appendix A. Some conclusions were obtained from the geospatial information presented in the webGIS. Figure 1 presents the spatial representation of the municipalities’ SVI referring to the basic education schools where participating teachers worked and the location of the schools. It is possible to verify that schools located in municipalities with low or very low SVI predominate in the Central-West, Southeast, and South.

The overall crude national absence rates were 17.25% for voice symptoms and 14.93% for psychological symptoms. As for FU adjusted rates, the medians (Q1–Q3) for voice outcomes were 17.06 (13.11–24.6) and for psychological outcomes, 16.23 (13.72–19.44).

The FUs with the highest adjusted absence rates (>Q3) for voice (25.19–29.03) and psychological symptoms (20.13–34.23) were in the North and Northeast Regions (Figure 2 and Figure 3). Noticeably, Pernambuco, Amazonas, Rio Grande do Norte, and Roraima present the highest rates in both outcomes. In the Southeast, São Paulo was the only FU with adjusted absence rates for psychological symptoms above Q3. It was also the FU with the highest adjusted absence rate for voice symptoms in the Region. 

The multivariate multilevel analysis model after the adjustment with all variables showed that absences due to voice and psychological symptoms had statistically significant associations respectively with high/very high SVI and intermediate and high/very high SVI. There was statistical significance with all adjusted variables in the multivariate multilevel model (Table 2). 

Absence from work for voice symptoms was associated with being a woman, younger than 34 years, with a working week longer than 40 h, having more than 20 years teaching, having income higher than three minimum wages, working in more than one school (including private schools), having more than one paid occupation, working in schools with 11 to 30 teachers, working in schools with intense noise, students’ indiscipline, longer commute, having experienced verbal and physical violence at school, excessive workload, few opportunities to learn new things, insufficient time to finish tasks, restricted autonomy, and low social support (Table 2).

The following factors associated with the psychological outcome differed from those in voice outcome: being older, having an income lower than three minimum wages, not working in more than one school, working in municipal and federal schools, and in schools with fewer than 10 teachers. The other precarious working conditions have a positive statistical significance in both outcomes (Table 2).

Model variance parameters measured by ICC indicate that 7% and 10% of the variations in absence rates, respectively for voice and psychological symptoms, were ascribable to differences between FUs not explained by the adjustment model. MOR translates the median increase in the odds of teachers being absent from work if they taught in a school in a different FU whose risk of absences is higher. The results indicate that this increase is 63% (MOR = 1.63) and 79% (MOR = 1.79) respectively for the voice and psychological outcomes (Table 2).

Including working conditions in the multivariate model improved the prediction of absences from work in both outcomes. The teachers who were absent for voice symptoms (vs. those who were not absent) were better discriminated against (0.71 = good model) than those who were absent for psychological symptoms (0.64 = average model) (Table 1). Hence, it is concluded that the adjustment variables are determinants in both outcomes. Including them in the multivariate model did not affect the ICC, indicating that the contextual effect of the FUs must be mediated by FU characteristics not considered in this study.

## 4. Discussion

This study—with a sample representative of the Brazilian urban school teachers registered in the 2014 Census—analyzed absences from work for voice and psychological symptoms, with three main findings: (i) the highest rates in both outcomes were concentrated in the North and Northeast regions of the country; nevertheless, the context of each FU must be analyzed; (ii) there was a relationship between social vulnerability, sex, and different working conditions with the teacher’s decision to be absent due to either outcome; (iii) precarious working conditions (perceived by the teachers) helped to explain the relationship between absences due to health problems and the social vulnerability of the municipality where the schools are located.

The higher rates in both outcomes in the North and Northeast regions in this study coincide with the municipalities with the greatest vulnerability to social exclusion, according to 2010 Ipea data [22]. On the other hand, although the Southeast had lower SVI [22], the municipality of São Paulo (the capital of the most populous FU in Brazil) had the highest absence rates for voice and psychological problems in the region. There is limited comparability of results between FUs with the literature because the continental size of the country hinders research with more than one FU. Thus, there were indications that health inequalities due to political, social, economic, and cultural contexts in each region should be considered when promoting teachers’ better voice and psychological health.

It was verified that working in schools in regions with high/very high SVI increased the odds of being absent for voice symptoms. As for absences for psychological symptoms, the association was in the opposite direction from the expected—i.e., teachers working in schools with high/very high SVI were less likely to be absent from school for this reason. It was also found that those who worked in schools with intermediate SVI were more likely to be absent than those who worked in schools with low/very low SVI. It has been identified that teachers in Brazil work even when they are sick because they lack the time to go to a doctor, are committed to students, avoid being downgraded due to poor attendance, and fear losing their jobs [7]. So, teachers who work in more vulnerable regions may have greater difficulties accessing health services, not recognize psychological symptoms, or suffer greater pressure not to be absent from work due to career consequences [3]. Since the analysis model discriminated better the teachers who were absent for voice symptoms than for psychological symptoms, it can be said that unfavorable socioeconomic conditions in the school surroundings contribute to teachers’ sick leaves.

Comparing our data from Brazil with those from high-income countries reveals a series of similarities. A prospective study in Finland showed that teachers who worked and lived in socioeconomically disadvantaged neighborhoods were on sick leave for long periods [6]. The authors point out as a limitation the little difference in social exclusion between Finnish school neighborhoods, as the education in this country has exemplary effectiveness in comparison with many other countries. Socioeconomically disadvantaged school neighborhoods were also related to French teachers’ high depersonalization and emotional exhaustion [20]. Greater absence from school was also found in Canadian children who studied in high-income-inequality neighborhoods, impairing their school achievements [39]. However, the teachers’ physical and mental health is not unrelated to the various macro aspects of the overall work-related social structure, which have helped increase sick leaves.

The SVI used in this study, which is calculated every 10 years, indicates the absence or insufficiency of greater assets, such as infrastructure, human capital, employment, and income for inhabitants in municipalities. The 2010 Ipea data showed that regional differences were decreasing in the last decades, diminishing the number of more vulnerable regions [22]. However, due to the COVID-19 pandemic, more recent municipal SVI indicators (not yet available) are expected to increase—i.e., indicate people’s poorer health and living conditions. Teachers’ mental health problems are still the main diagnoses that justify sick leaves [11]. A study verified that during the pandemic, 82.3% of teachers had at least one mental health problem [40]. The population’s aggravated health without greater financial investments to improve schools may have direct effects on teachers’ working conditions and income, which may pose a risk to the quality of education. It should be noted that the present study approaches data prior to the pandemic (2015–2016), which underscores the recent deterioration of working conditions in the education area.

Altogether, 83.3% of the teachers in this study are women. In both outcomes, being a woman increased the odds of being absent from work. In comparison with men, women are at a 36% higher risk of absences from work due to vocal symptoms and 22% higher risk due to psychological symptoms; all other covariates were equal and in the same FU. There is scientific evidence of greater absenteeism for various health reasons among women [41,42], which can be explained by their greater exposure to stressful factors in school occupations that do not require as much qualification. A study demonstrated that high stress, high work pace, great emotional demands, low influence on work, and precarious psychosocial security predominated in female teachers, in contrast with male ones [43]. The work differences between teachers of both sexes must be considered, and the analysis must include work done away from the workplace, such as household chores [44].

Teachers who reported working more than 40 h a week and being in the occupation for longer were also more likely to be absent for voice and psychological symptoms. A systematic review concluded that working more than 40 h a week is associated with a depressive state, anxiety, sleep status, and coronary cardiac disease [45]. A study based on machine learning algorithms showed that although the workload in hours contributed to public teachers’ absence for morbidities in some analysis models, the best model pointed to the time of employment as one of the predictive variables [46]. It can be said that longer working weeks and continuous exposure to inadequate working conditions throughout the teachers’ careers can cause greater physical and mental exhaustion, thus predisposing them to be on sick leave.

It was verified that as teachers grow older, they are less likely to be absent from work for voice symptoms, whereas absences for psychological symptoms are more likely to occur in older teachers. A systematic review indicates that absenteeism is usually due to diseases, rather than age alone [47]. Teachers deny or minimize sickness to take care of others, and only pay attention to their condition when the problem intensifies [44].

The literature shows that absences for voice symptoms are mostly short [30], whereas those for psychological symptoms tend to last longer [14]. A study showed that long-term absence from work occurred more often in older teachers [13]. A systematic review reports that Brazilian public school teachers have high levels of emotional exhaustion in combination with a prevalence of burnout—nonetheless, they are persistently motivated and idealist [48], which may help them continue working, despite being sick. Voice symptoms are better recognized by others because they are physical and restrict or limit communication in the classroom, unlike psychological symptoms, which are often neglected or not legitimated. Hence, younger teachers with voice symptoms probably are absent from work to recover their voice and develop strategies to self-regulate it over time. On the other hand, older teachers with more chronic psychological symptoms developed over time may be absent from classrooms for longer and more often to recover.

The results of this study showed that the perception of having to raise the voice because of the intense noise and having experienced physical violence at school were factors that increased by 50% or more the odds of being absent for voice and psychological symptoms. A study showed that the cumulative effects of noise on teachers, especially by the end of the working week, overlap with vocal and mental fatigue [49]. There is evidence of the relationship between intense noise at school due to students’ indiscipline, experiencing verbal aggression, heavy workload, working in more than one teaching modality, and in schools with more than 30 teachers [50]. In Denmark, teachers who worked in classrooms with worse acoustics perceived their social setting as more competitive, burdened with conflicts, and less relaxed and comfortable, making it a reason for the greater intention to leave work [51]. Basic education teachers in Southern Brazil public schools who suffered physical aggression were absent more often from work [11]. A Canadian study showed that teachers who suffered violence at work had physical and mental effects and reacted with fear, anguish, and isolation [52]. Students take to school traces of violence they face in the community [21], and teachers are often unprepared to deal with adverse events at school.

Positive associations between the other precarious working conditions used in analysis model adjustment and the outcomes agree with the results of other studies [9,10,11]. Stressful factors, such as violence at school, relationship difficulties at work, restricted autonomy, little possibility of creative activities, lack of time to correct homework and tests, poor overall working conditions, and constant political/educational changes affect the teachers’ health [21]. Thus, inefficient work organization at school is related to increased absences of teachers for voice and psychological symptoms.

The teachers in this study who were absent from work for psychological symptoms worked in only one school, with fewer teachers, and had a lower income. This can be explained by the possible reverse causality in the cross-sectional design of this study. It can be feasibly considered that their mental health status led classroom teachers to adjust their employment to a less overloaded situation, though with a lower income.

On the other hand, it has been proved that teachers who work in more than one school, in schools with more teachers, and consequently had a higher income are absent from work more often for voice symptoms. A previous study highlights the influence of dissatisfaction with employment conditions, organizational policies, wages, occupational safety, interpersonal relationships, and physical aspects of the workplace as determinants of absenteeism, in contrast with individual issues [3]. However, the combination of the various work-related factors changes within and between Brazilian FUs and can either favor or not healthy work environments in schools, influencing teachers’ sickness and sick leaves.

This study has some limitations. Since it is cross-sectional, reverse causality cannot be ruled out. Nevertheless, teachers who perceive more precarious working conditions and work in more socially vulnerable regions are probably the ones who are absent more often for health problems. Secondly, the study did not include teachers who worked in rural schools because SVI is not calculated for rural areas. Finally, this survey gathered no information on the number of sick leave spells during the previous 12 months, which would have contributed to a greater discussion on the differences found between outcomes.

The study has various strengths as well. It analyzed a probabilistic and representative sample of Brazilian urban schoolteachers and presented the adjusted absence rates for voice and psychological symptoms of each of the 27 FUs. Moreover, it led to more in-depth knowledge of the teachers’ self-perception of working conditions, which is important to mediate the relationship between sick leaves and social vulnerability in metropolitan schools. The teachers’ self-perception of health belongs and is connected to a social environment, which must be considered. Regardless of individual factors that influence their decision to be absent from work, precarious working conditions are strongly associated with the decision to take sick leave. Precarious working conditions cannot be seen as normal or unchangeable. Valuing teachers by giving them better working conditions would help improve their health and education indicators. Besides improving school infrastructure, teachers and students should be heard and encouraged to discuss their practices and difficulties and how to solve them, thus improving their performance and general skills [7]. Future studies should consider both longitudinal and qualitative designs.

Furthermore, a user-friendly webGIS was developed, allowing users to dynamically visualize and analyze data related to the schools and the SVI index per municipality. Data in this webGIS are available to all users, with tools that can help to analyze its information. The webGIS provides free access to any user and can help with management decision-making.

## 5. Conclusions

The results confirm the relationship between precarious working conditions and higher absence rates for voice and psychological symptoms, reinforcing the need for investments to improve teachers’ working conditions. We highlight the priority of reducing noise in the classroom and ensuring greater security in schools to mitigate violence, considering the reality of each school as pointed out by teachers and students. Focusing on improving schools in municipalities whose SVI is intermediate or high/very high and in FUs with the highest absence rates can favor the teachers’ voice and psychological health. Multiprofessional measures to promote, prevent, and rehabilitate teachers’ health should be analyzed and potentialized.

## Figures and Tables

**Figure 1 ijerph-20-02972-f001:**
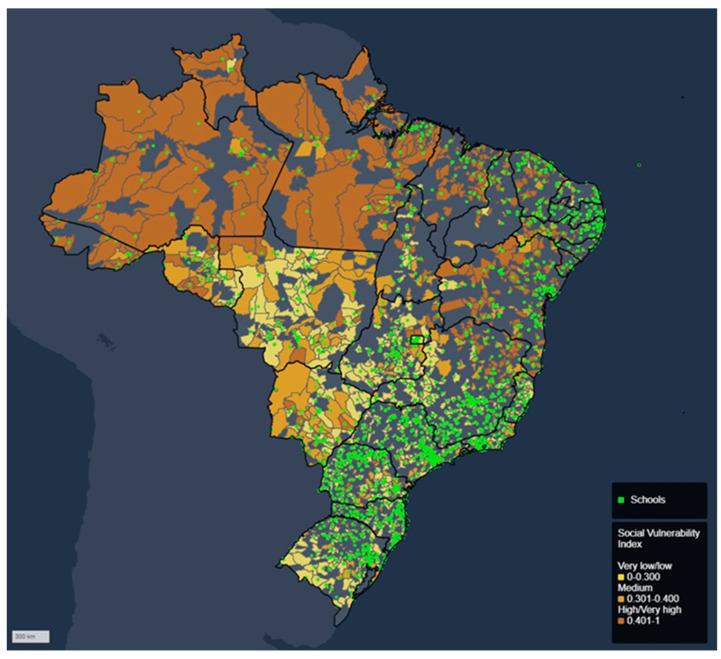
Spatial representation of metropolitan schools and the SVI of Brazilian municipalities.

**Figure 2 ijerph-20-02972-f002:**
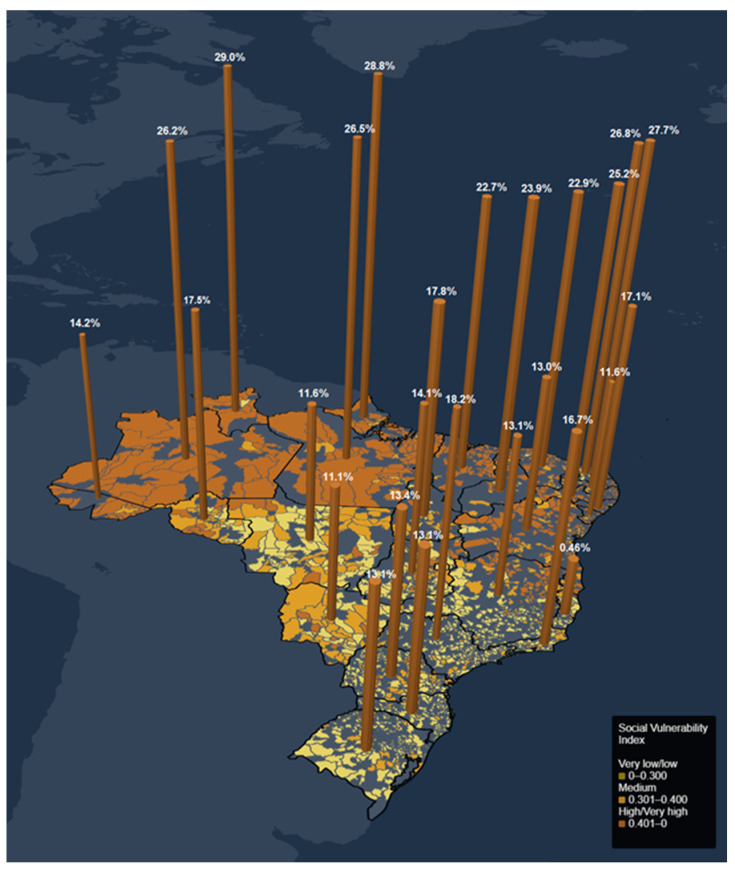
Adjusted rates of teachers’ absences from work due to vocal symptoms for each FU in Brazil (27 states and the Federal District) and the SVI of municipalities.

**Figure 3 ijerph-20-02972-f003:**
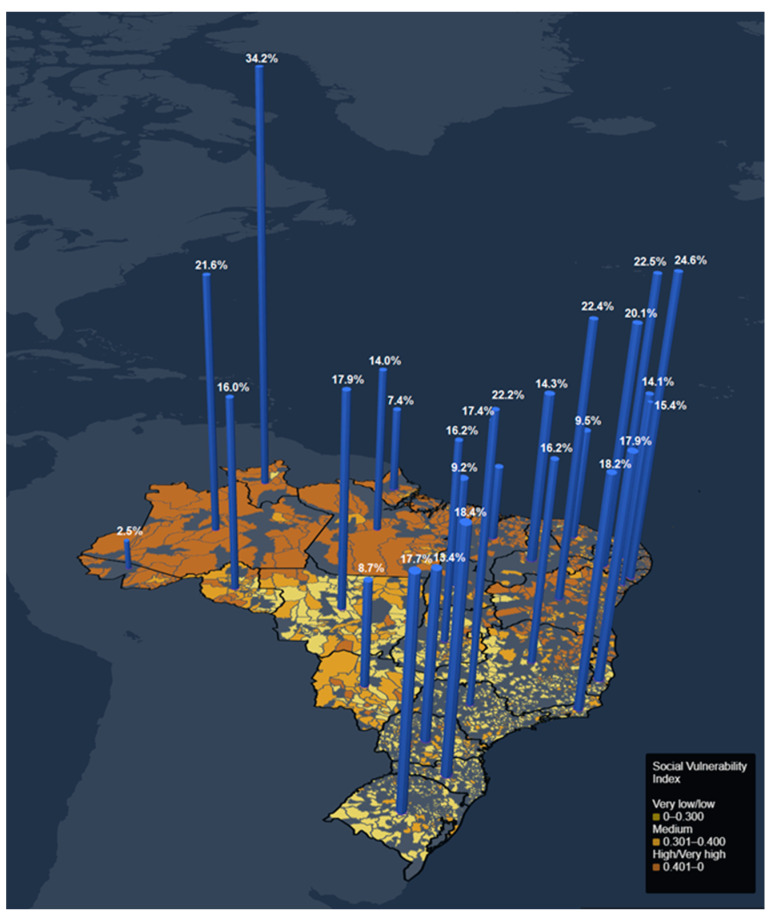
Adjusted rates of teachers’ absences from work due to psychological symptoms for each FU in Brazil (27 states and the Federal District) and the SVI of municipalities.

**Table 1 ijerph-20-02972-t001:** Municipality SVI ranges—Ipea 2015.

SVI	Range
Very low social vulnerability	from 0 to 0.200
Low social vulnerability	from 0.201 to 0.300
Intermediate social vulnerability	from 0.301 to 0.400
High social vulnerability	from 0.401 to 0.500
Very high social vulnerability	from 0.501 to 1

**Table 2 ijerph-20-02972-t002:** Multivariate association of the national rate of absences due to vocal and psychological symptoms with the SVI of Brazilian municipalities adjusted for sex, age, and working conditions reported by teachers.

Features		Absences Due to Vocal Symptoms (%)	OR (95% CI)	Absences Due to Psychological Symptoms (%)	OR (95% CI)
Intercept			0.04 (0.03; 0.04) *		0.01 (0.01; 0.01) *
Sex	Male	16.7	1	16.5	1
	Female	83.3	1.36 (1.35; 1.38) *	83.5	1.22 (1.21; 124) *
Age (years)	≤34	35.6	1	27.0	1
	35–44	30.3	0.82 (0.81; 0.82) *	31.6	1.11 (1.09; 1.12) *
	45–54	24.9	0.63 (0.62; 0.64) *	29.4	1.01 (0.99; 1.02)
	≥55	9.2	0.65 (0.64; 0.66) *	12.0	1.22 (1.20; 1.24) *
Working weeks longer than 40 h	No	37.7	1	36.5	1
	Yes	62.3	1.18 (1.17; 1.19) *	63.5	1.30 (1.29; 1.31) *
Employment time (years)	<10	33.3	1	26.9	1
	10–20	31.6	1.03 (1.02; 1.05) *	33.2	1.32 (1.31; 1.34) *
	>20	35.0	1.34 (1.32; 1.36) *	39.9	1.52 (1.49; 1.54) *
Income from school in minimum wages	>three	35.7	1	34.9	1
	Up to three	64.3	0.81 (0.80; 0.81) *	65.1	1.06 (1.05; 1.07) *
Working in more than one school	No	42.8	1	48.5	1
	Yes	57.2	1.34 (1.32; 1.35) *	51.5	0.89 (0.88; 0.90) *
Working in another activity	No	89.5	1	89.8	1
	Yes	10.5	1.10 (1.09; 1.12) *	10.2	1.04 (1.02; 1.05) *
School administration	Private	19.7	1	12.9	1
	Municipal	45.1	0.99 (0.98; 1.00)	52.1	1.29 (1.26; 1.32) *
	State	32.3	0.99 (0.98; 1.00)	31.6	1.97 (1.94; 2.00)
	Federal	2.9	0.46 (0.45; 0.48) *	3.4	1.58 (1.55; 1.60) *
School size by the number of teachers	≤10	5.2	1	5.2	1
	11–20	13.6	0.83 (0.82; 0.85) *	16.9	0.97 (0.95; 0.99) **
	21–30	20.6	1.18 (1.16; 1.20) *	16.7	0.74 (0.72; 0.75) **
	>30	60.6	0.99 (0.97; 1.01)	61.2	0.75 (0.73; 0.76) **
Perception of intense noise at school	Never or almost never/Rarely	18.3	1	16.7	1
	Often/Sometimes	81.7	1.92 (1.90; 1.94) *	83.3	1.86 (1.84; 1.88) **
Student indiscipline	Never or almost never/Rarely	17.0	1	16.2	1
	Often/Sometimes	83.0	1.32 (1.31; 1.34) *	83.8	1.25 (1.23; 1.26) *
Roundtrip home-work commute (in minutes)	10–20	30.6	1	30.6	1
	21–50	34.5	1.33 (1.32; 1.35) *	37.2	1.31 (1.30; 1.32) *
	>51	34.9	1.40 (1.38; 1.41) *	32.2	1.17 (1.15; 1.18) *
Experiencing verbal violence in school	Never	56.6	1	49.5	1
	Once/twice or more	43.4	1.35 (1.33; 1.36) *	50.5	1.75 (1.74; 1.77) *
Experiencing physical violence in school	Never	94.5	1	93.5	1
	Once/twice or more	5.5	1.57 (1.54; 1.60) *	6.5	1.50 (1.47; 1.53) *
Excessive workload	Never or almost never/Rarely	9.8	1	9.6	1
	Often/Sometimes	90.2	1.66 (1.64; 1.68) *	90.4	1.39 (1.37; 1.41) *
Few opportunities to learn new things	Never or almost never/Rarely	84.5	1	78.5	1
	Often/Sometimes	15.5	1.01 (1.00; 1.03) **	21.5	1.44 (1.42; 1.46) *
Insufficient time to complete tasks	Never or almost never/Rarely	77.9	1	70.3	1
	Often/Sometimes	22.1	1.46 (1.44; 1.47) *	29.7	2.23 (2.21; 2.26) *
Limited autonomy	Never or almost never/Rarely	79.2	1	75.9	1
	Often/Sometimes	20.8	1.16 (1.14; 1.17) *	24.1	1.33 (1.31; 1.34) *
Low social support	Never or almost never/Rarely	48.6	1	42.3	1
	Often/Sometimes	51.4	1.19 (1.18; 1.20) *	57.7	1.42 (1.41; 1.44) *
SVI of the municipalities	Low	54.5	1	59.0	1
	Median	30.9	1.01 (1.00; 1.02)	31.5	1.15 (1.13; 1.16) *
	High	14.6	1.05 (1.03; 1.07) *	9.5	0.86 (0.85 0.88) *
Variance parameters		Estimate		Estimate	
Variance FU		0.2655		0.3754	
ICC		0.07		0.10	
MOR		1.63		1.79	
c-statistic		0.71		0.64	

SVI = Social Vulnerability Index; OR = odds ratio; CI = Confidence interval; FU = federative unit; ICC = Intraclass correlation coefficient (variance/(variance + π^2^/3)); MOR = Median chance ratio (exp [0.6745 × √(2 * variance)]); * *p*-value ≤ 0.001; ** *p*-value ≤ 0.05.

## Data Availability

The raw data supporting the conclusions of this article will be made available by the corresponding author upon reasonable request. Data is not publicly available due to involving humans.

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
