# Peer review of "Social Vulnerability of Brazilian Metropolitan Schools and Teachers’ Absence from Work Due to Vocal and Psychological Symptoms: A Multilevel Analysis"

_ijerph, 2023, doi:10.3390/ijerph20042972_

Round 1

Reviewer 1 Report

This is a manuscript that asess the variables associated with teacher absences in Brazil. 

I believe that this is a commendable work and I have a few comments that I hope improve the overall manucript.

Figure 2 and 3 are complicated to read, also, the use of 3-d make them confusing. Please consider changing them substanstially. 

Please also, consider including the especification of the model or the models that were tested (the formula). This could be helpful to make more clear what was included in your models. 

Did you test different structures in random effects? If this is not the case, I strongly recommend to support your decision based on relevant literature

Reviewer 2 Report

Social vulnerability of Brazilian metropolitan schools and teachers' absence from work due to vocal and psychological symptoms: a multilevel analysis

Summary: The authors worked on ~5000 randomly selected teachers from 26 Brazilian state and federal districts to determine the relationship between teachers’ absence rate due to voice and psychological symptoms. The authors found that 17.25% were absent due to voice symptoms, and 14.93% were due to psychological symptoms. A positive relationship between voice outcome and high social vulnerability index, whereas psychological symptoms were negatively associated. Women working in socially vulnerable regions with schools in precarious conditions had higher chances of being absent due to voice or psychological conditions. Based on the current findings, the authors conclude that more investment is required in schools to improve working conditions in schools.

Comments on abstract:  

From the abstract, the study appears very impactful and potentially may have a widespread audience. But quantitative details are missing for multiple claims.

1.      The abstracts lack information on the data collected on randomly sampled teachers and the type of multivariate analysis performed.

2.      The abstracts have strong claims about female absentee teachers but don’t mention what % of females are teachers in the total sample.

3.      While the claims appear strong, it is unclear whether they are statistically significant. Information on statistical analysis is missing.

4.       Being a woman and working in schools with precarious conditions and in…” It is unclear how the authors reach to this conclusion.

Comment on Dataset:

1.      The data set is from 2015-2016 survey, which is reasonably old considering the latest changes and changes related to COVID-19. How do the authors explain the validity of the results?

2.      SVI has been given fair amount of importance but was only collected for urban areas. This information must be mentioned in the abstract, otherwise, the results are misleading. The manuscripts mention that the SVI is collected every 10 years. What is the rationale behind using the 2010 data, when 2020 data might be available?

3.      The authors mention that they have used a set of the survey questionnaires, but the questionnaires are missing in the manuscript. It is strongly suggested to mention the questionnaire in the manuscript or as additional material.

4.      When was the survey conducted?

Comment on Results:

5.      “after the multivariate model was adjusted for sex, age, and working conditions”.. What kind of adjustment was done concerning sex, age, and working conditions?

6.      “The FUs with the highest absence rates (> Q3) for voice and psychological sym” .. It is recommended to mention the absence rate instead of saying high absent rate.

7.      How was the absent rate calculated? Is it based on national absent rate calculation or were regional evaluations done?

8.      “Absence from work for voice symptoms was associated with being a woman (“.. which this statement might have an impact, without quantified results, the insight is inconclusive. For example, what are the numbers, are they significantly different from other women in the group, are they different from male counter parts? Is is driven be regions or overall country gave similar effect? What test are performed to evaluate the effect? Tests mentioned  in table 2 are sufficient but doesn’t explain multivariate interaction.

Comments on Discussion:

9.      “This study – with a sample representative of the 2,229,269 Brazilian”.. This study does not represent Brazilian teachers, as it focuses on teachers only from urban areas. The statement in the manuscript is entirely misleading.

10.   working in schools in municipalities with greater social vulnerability, being a woman, and working in precarious conditions increased the odds of being absent for both reasons and;    >>> This statement is misleading.

11.   the working conditions perceived by teachers were deter-minant to explain the relationship between absences for health issues and the social vul-nerability in the school surroundings. >>> Coincidence should be termed as causality.

12.   Thus, interventions to promote teachers’ better voice and psychological health must give priority to health inequities due to the political, social, economic, and cultural reality of each region.  >>> May be a good point but lacks significant evidence.

13.   Teach-ers who work in more vulnerable regions may have greater difficulties accessing health services, less recognition of psychological symptoms, or greater pressure not to be absent from work because of financial consequences, such as lower income.  >>> Did the authors find any evidence of low income from their own data or any reference? Otherwise, it is mere speculation.

14.   In both outcomes, being a woman increased the odds of being absent from work. >>>> The study never mentioned how many women teachers were included in the survey. It may so happen that the proportion of men were see creating a bias in the results.

The conclusions of the study are pretty strong and questionable and often lack appropriate evidence.

Reviewer 3 Report

The research study described in the manuscript adopted an innovative and practical application of using GIS for identifying patterns in teacher absence caused by voice and emotional problems. The use of GIS in educational research is relatively new, and the results reported in the manuscript can stimulate new research using the GIS technology. 

The diagrams included in the manuscript are very useful, especially they can be viewed dynamically.

Given the rich findings, the provision of only one suggestions of improving working conditions seems weak. More concrete recommendations in terms of policy and pedagogy can be added.

Round 2

Reviewer 2 Report

The editing of the manuscript is acceptable. There are several spelling mistakes present in the article. Please ensure the spell check is done before accepting the article.

Author Response

The article was revised by a professional translator fluent in
English and Portuguese, who edited the article for language, grammar, punctuation, spelling, and general style, without altering its content.
